# Novelty Search in representational space for sample efficient exploration

## Abstract

We present a new approach for efficient exploration which leverages a low-dimensional encoding of the environment learned with a combination of model-based and model-free objectives. Our approach uses intrinsic rewards that are based on a weighted distance of nearest neighbors in the low dimensional representational space to gauge novelty. We then leverage these intrinsic rewards for sample-efficient exploration with planning routines in representational space. One key element of our approach is that we perform more gradient steps in-between every environment step in order to ensure the model accuracy. We test our approach on a number of maze tasks, as well as a control problem and show that our exploration approach is more sample-efficient compared to strong baselines.

## 1 Introduction

In order to solve a task efficiently in Reinforcement Learning (RL), one of the main challenges is to gather informative experiences thanks to an efficient exploration of the state space. A common approach to exploration is intrinsic rewards correlated with some novelty heuristics (Schmidhuber, 2010; Stadie et al., 2015; Houthooft et al., 2016). With intrinsic rewards, an agent can be incentivized to efficiently explore its state space. A direct approach to calculating these novelty heuristics is to derive a reward based on the observations, such as a count-based reward (Bellemare et al., 2016; Ostrovski et al., 2017) or a prediction-error based reward (Burda et al., 2018b). However, an issue occurs when measuring novelty directly from the raw observations, as some information in the pixel space (such as randomness) might be irrelevant. In this case, if an agent wants to efficiently explore its state space it should only focus on meaningful and novel information.

In this work, we propose a method of sample-efficient exploration by leveraging novelty heuristics in a meaningful abstract state space. We leverage a low-dimensional abstract representation of states, which is learned by fitting both model-based and model-free components through a joint representation. This provides a meaningful abstract representation where states that are close temporally in *dynamics* are brought close together in low-dimensional representation space. We also add additional constraints to ensure that a measure of distance between states is meaningful. With this distance in representational space, we form a novelty heuristic inspired by the Novelty Search algorithm (Lehman and Stanley, 2011) to generate intrinsic rewards that we use for efficient exploration. We show that with a good low-dimensional representation of states, a policy based on planning with our novelty heuristic is able to explore with high sample-efficiency.

In our experiments, we measure the effectiveness of our exploration methods by the number of samples required to explore the state space. One key element of our approach is that we perform more gradient steps in-between every environment step in order to ensure the model accuracy is high (and hence ensure an accurate novelty heuristic). Through this training scheme, our agent is also able to learn a meaningful representation of its state space in an extremely sample-efficient manner.

## 2 Problem formulation

Our agent interacts with its environment over discrete time steps, modeled as a Markov Decision Process (MDP), defined by the 6-tuple $(\mathcal{S}, \mathcal{S}_0, \mathcal{A}, \tau, \mathcal{R}, \mathcal{G})$ (Puterman, 1994). In this setting, $\mathcal{S}$ is the state space, $\mathcal{S}_0$ is the initial state distribution, $\mathcal{A}$ is the discrete action space, $\tau : \mathcal{S} \times \mathcal{A} \to \mathcal{S}$ is

the transition function that is assumed deterministic (with the possibility of extension to stochastic environments with generative methods), $R : \mathcal{S} \times \mathcal{A} \to \mathcal{R}$ is the reward function ($\mathcal{R} = [-1, 1]$), $\mathcal{G} : \mathcal{S} \times \mathcal{A} \to [0, 1)$ is the per time-step discount factor. At time step $t$ in state $s_t \in \mathcal{S}$, the agent chooses an action $a_t \in \mathcal{A}$ based on policy $\pi : \mathcal{S} \times \mathcal{A} \to [0, 1]$, such that $a_t \sim \pi(s_t, \cdot)$. After taking $a_t$, the agent is in state $s_{t+1} = \tau(s_t, a_t)$ and receives reward $r_t \sim R(s_t, a_t)$ and a discount factor $\gamma_t \sim \mathcal{G}(s_t, a_t)$. Over $n$ environment steps, we define the history of visited states as $H_s = (s_1, \ldots, s_n)$, where $s_i \in \mathcal{S} \; \forall i \in \mathbb{N}$. In RL, the usual objective is to maximize the sum of expected future rewards $V^\pi(s) = \mathbb{E}\left[ r_t + \sum_{i=1}^{\infty} \left( \prod_{j=0}^{i-1} \gamma_{t+j} \right) r_{t+i} | s = s_t, \pi \right]$.

To learn a policy $\pi$ that maximizes the expected return, an RL agent has to efficiently explore its environment. In this paper, we consider exploration tasks with sparse rewards, and are interested in exploration strategies that require as few steps as possible to explore the state space.

## 3 ABSTRACT STATE REPRESENTATIONS

In this work, we focus on learning a lower-dimensional representation of state when our state (or observations in the partially observable (Kaelbling et al., 1998) case) is high-dimensional (Dayan, 1993; Tamar et al., 2016; Silver et al., 2016; Oh et al., 2017; de Bruin et al., 2018; Ha and Schmidhuber, 2018; François-Lavet et al., 2018; Hafner et al., 2018; Gelada et al., 2019). We learn our lower-dimensional abstract representation by encoding the high-dimensional state with an encoder $\hat{e} : \mathcal{S} \to \mathcal{X}$ parameterized by $\theta_{\hat{e}}$, where $\mathcal{X} \in \mathbb{R}^{n_\mathcal{X}}$. The dynamics are learned via the following functions: a transition function $\hat{\tau} : \mathcal{X} \times A \to \mathcal{X}$ parameterized by $\theta_{\hat{\tau}}$, a reward function $\hat{r} : \mathcal{X} \times A \to [0, R_{max}]$ parameterized by $\theta_{\hat{r}}$, and a per timestep discount factor function $\hat{\gamma} : \mathcal{X} \times A \to [0, 1)$ parameterized by $\theta_{\hat{\gamma}}$.

In order to leverage all past experiences, we use an off-policy learning algorithm that sample transition tuples $(s, a, r, \gamma, s')$ from a replay buffer. We first encode our current and next states:

$$x \leftarrow \hat{e}(s; \theta_{\hat{e}}), \; x' \leftarrow \hat{e}(s'; \theta_{\hat{e}}),$$

where $x, x' \in \mathcal{X}$ and $\theta_{\hat{e}}$ are our encoder parameters. The Q-function is learned using the DDQN (van Hasselt et al., 2015) algorithm, which uses the target

$$Y = r + \gamma Q(\hat{e}(s'; \theta_{\hat{e}^-}), \operatorname*{argmax}_{a' \in \mathcal{A}} Q(x', a'; \theta_Q); \theta_{Q^-}), \tag{1}$$

where $\theta_{Q^-}$ and $\theta_{\hat{e}^-}$ are parameters of an earlier buffered Q-function (or our target Q-function) and encoder respectively. The agent then minimizes the following loss:

$$L_Q(\theta_Q) = (Q(x, a; \theta_Q) - Y)^2. \tag{2}$$

We learn the dynamics of our environment through the following losses:

$$L_R(\theta_{\hat{e}}, \theta_{\hat{r}}) = |r - \hat{r}(x, a; \theta_{\hat{r}})|^2, \tag{3a}$$

$$L_\mathcal{G}(\theta_{\hat{e}}, \theta_{\hat{\gamma}}) = |\gamma - \hat{\gamma}(x, a; \theta_{\hat{\gamma}})|^2, \tag{3b}$$

$$L_\tau(\theta_{\hat{e}}, \theta_{\hat{\tau}}) = |[x + \hat{\tau}(x, a; \theta_{\hat{\tau}})] - x'|^2, \tag{3c}$$

By jointly learning the weights of the encoder and the different components, the abstract representation is shaped in a meaningful way according to the dynamics of the environment. For instance, states that are temporally close through the transition function are brought close in the lower-dimensional space–a phenomenon reported in François-Lavet et al. (2018). To ensure that learning the transition function between abstract states doesn't tend to collapse the abstract representations to a constant, we add the following soft constraints on the entropy:

$$L_{d1}(\theta_{\hat{e}}) = exp(-C_d||\hat{e}(s; \theta_{\hat{e}}) - \hat{e}(s'; \theta_{\hat{e}})||_2), \tag{4}$$

where $s, s'$ are two randomly sampled states from the agent's history and $C_d$ is a hyperparameter. We also use a soft constraint between consecutive states that tends to enforce that two consecutive abstract representations are at a distance of at least $\omega$:

$$L_{consec}(\theta_{\hat{e}}) = max(||\hat{e}(s_1; \theta_e) - \hat{e}(s_2; \theta_e)||_2 - \omega, 0), \tag{5}$$

where $\omega$ is a hyperparameter (a discussion of this loss is provided in Appendix A). The hyperparameter $\omega$ can be used to estimate the accuracy of our transition loss, hence of our novelty estimates.

In order to gauge when our representation is accurate enough to use our novelty heuristic, we use a function of this hyperparameter and transition loss to set a cutoff point for accuracy to know when to take the next environment step. If $\omega$ is the minimum distance between successive states, then when $L_\tau \leq \left(\frac{\omega}{\delta}\right)^2$, the transitions are on average within a ball of radius $\frac{\omega}{\delta}$ of the target state. Here $\delta > 1$ is a hyperparameter that we call the slack ratio. Before taking a new step in the environment, we keep training all the parameters with all these losses until this threshold is reached and our novelty heuristic becomes useful. Details on the slack ratios used in the experiments are given in Appendix E.

We minimize the sum of all the aforementioned losses through gradient descent with learning rate $\alpha$:

$$\mathcal{L} = L_R(\theta_{\hat{e}}, \theta_{\hat{r}}) + L_{\mathcal{G}}(\theta_{\hat{e}}, \theta_{\hat{\gamma}}) + L_\tau(\theta_{\hat{e}}, \theta_{\hat{\tau}}) + L_Q(\theta_Q) + L_{d1}(\theta_{\hat{e}}) + L_{consec}(\theta_{\hat{e}}). \quad (6)$$

Through these losses, the agent learns a low-dimensional representation of the environment. These losses are well-suited for our novelty metric because of the meaningfulness the losses ensure in terms of the $L2$ norm in our representation space (see Appendix C). Planning techniques that combine the knowledge of the model and the value function can then be used to select actions that will maximize intrinsic rewards.

## 4 COMBINING MODEL-FREE AND MODEL-BASED COMPONENTS FOR EXPLORATION POLICIES

Similarly to previous works (e.g. Oh et al., 2017), we use a combination of model-based planning with model-free Q-learning to obtain a good policy. For our approach, we calculate rollout estimates of next states based on our transition model $\hat{\tau}$ and sum up the corresponding rewards (which we denote as $r : \mathcal{X} \times A \to [0, R_{max}]$ and can be a combination of both intrinsic and extrinsic rewards). We calculate expected returns based on the discounted rewards of our $d$-depth rollouts:

$$\hat{Q}^d(x, a) = \begin{cases} r(x, a) + \hat{\gamma}(x, a; \theta_{\hat{\gamma}}) \max_{a' \in \mathcal{A}} \hat{Q}^{d-1}(\tau(x, a'; \theta_{\hat{\tau}}), a'; \theta_Q), & \text{if } d > 0 \\ Q(x, a; \theta_Q), & \text{if } d = 0 \end{cases} \quad (7)$$

Note that we simulate only $b$-best options at each expansion step based on $Q(x, a; \theta_Q)$, where $b \leq |\mathcal{A}|$.

We then use a simple sum of the Q-values obtained with planning up to a depth $D$:

$$Q_{plan}^D(x, a) = \sum_{d=0}^{D} \hat{Q}^d(x, a).$$

The estimated optimal action is given by $\underset{a \in \mathcal{A}}{\operatorname{argmax}} \, Q_{plan}^D(x, a)$. The actual action chosen at each step follows an $\epsilon$-greedy strategy ($\epsilon \in [0, 1]$), where the agent follows the estimated optimal action with probability $1 - \epsilon$ and a random action with probability $\epsilon$.

## 5 NOVELTY SEARCH IN ABSTRACT REPRESENTATIONAL SPACE

Our approach uses *intrinsic motivation* (Schmidhuber, 1990; Chentanez et al., 2005; Achiam and Sastry, 2017) where an agent rewards itself based purely on the fact that it gathers interesting experiences. The specific formulation of the rewards is inspired by the Novelty Search algorithm (Lehman and Stanley, 2011). Since the concept of proximity loses meaning in high-dimensional spaces (Aggarwal et al., 2002), the nearest neighbor problem is ill-defined. So instead of calculating a measure of novelty in pixel space, we leverage the low-dimensional abstract representations.

To get a novelty metric for our abstract representations, we can apply a weighted version of the novelty measure in abstract representational space to our current state $s$ with our encoder applied $\hat{x} \leftarrow \hat{e}(s; \theta_{\hat{e}})$, and a history of encoded states. We define the following novelty measure over abstract representational space:

$$\rho_{\mathcal{X}}\left(\hat{e}(s; \theta_{\hat{e}})\right) = \sum_{i=1}^{k} \frac{1}{i} dist\left(\hat{e}(s; \theta_{\hat{e}}), \hat{e}(s_i; \theta_{\hat{e}})\right), \quad (8)$$

where $s_i$ is the $i$th closest state to $s$ in the abstract representations and $dist$ is a distance measure in representational space. In this work, we use $\ell_2$ norm as our measure of distance. We motivate this choice in Appendix C In the heuristic above, we weight the distances between the $k$ nearest neighbors and $s$ by their ranking based on distances from the corresponding abstract states. We do this in order to compensate for biases towards state-action pairs that have fewer direct neighbors as opposed to purely novel states. We clarify this point with a toy example in Appendix B and we compare this formulation with an unweighted average in Appendix F. We also analyze the limiting behaviour of our novelty reward in Appendix D.

Incorporating all of the above, our algorithm is described in Algorithm 1.

---

**Algorithm 1:** The Novelty Search algorithm in abstract representational space.

---

1   **Initialization:** transition buffer $H$, agent policy $\pi$;
2   Sample $n_{init}$ initial random transitions, let $t = n_{init}$;
3   **while** $t \leq n_{max}$ **do**
      // We update our dynamics model and Q-function every $n_{freq}$ steps
4      **if** $t \mod n_{freq} == 0$ **then**
5         **while** $j \leq n_{iters}$ *or* $L_\tau \leq \left(\frac{\omega}{\delta}\right)^2$ **do**
6            Sample batch of transitions $(s, a, r_{extr}, r_{intr}, \gamma, s') \in H$;
7            Train dynamics model with $(s, a, r_{extr}, \gamma, s')$;
8            Train Q-function with $(s, a, r_{extr} + r_{intr}, \gamma, s')$;
9         **end**
10        $\forall(s, a, r_{extr}, r_{intr}, \gamma, s') \in H$, set $r_{intr} \leftarrow \rho_{\mathcal{X}}(\hat{e}(s'; \theta_{\hat{e}}))$;
11      **end**
12      $a_t \sim \pi(s_t)$;
13      Take action in environment: $s_{t+1} \leftarrow \tau(s_t, a_t), r_{t,extr} \leftarrow R(s_t, a_t), \gamma_t \leftarrow \mathcal{G}(s_t, a_t)$;
14      Calculate intrinsic reward: $r_{t,intr} \leftarrow \rho_{\mathcal{X}}(\hat{e}(s_{t+1}; \theta_{\hat{e}}))$
15      $H \leftarrow H \cup \{(s_t, a_t, r_{t,extr}, r_{t,intr}, \gamma_t, s_{t+1})\}$;
16   **end**

---

## 6   EXPERIMENTS

We conduct experiments on environments of varying difficulty. All experiments use a training scheme where parameters converge on an accurate representation before taking an environment step. We optimize the losses (over multiple training iterations) given in Section 3 before each environment step to train the agent's state representation. We discuss all environment-specific hyperparameters in Appendix E.

### 6.1   LABYRINTH EXPLORATION

We consider two $21 \times 21$ versions of the grid-world environment, as per Figure 1. The first is an open labyrinth grid-world, with no walls except for bordering walls. The second is a similar sized grid-world split into four connected rooms. In these environments the actions $\mathcal{A}$ are the four cardinal directions. These environments have no rewards or terminal states and the goal is to explore, agnostic of the task. We use two metrics to gauge exploration for this environment: the first is the ratio of states visited only once, the second is the proportion of total states visited.

#### 6.1.1   OPEN LABYRINTH

In the open labyrinth experiments (Figure 4a), we compare a number of variations of the above policy with a random baseline and a count-based baseline (Bellemare et al., 2016) (as we can count states in this tabular setting). Variations of the policy include an argmax over state values ($d = 0$) and planning depths of $d = \{1, 5\}$.

Our method is able to leverage a good learnt representation in order to explore in a sample-efficient way. All variations of our method outperform the two baselines in this task, with a slight increase

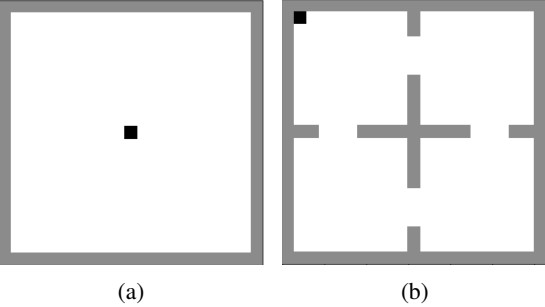

(a)                           (b)

Figure 1: *Left*: Open labyrinth - A $21 \times 21$ empty labyrinth environment. *Right*: 4-room labyrinth - A $21 \times 21$ 4-room labyrinth environment.

in performance as planning depth $d$ increases. In the open labyrinth, our agent is able to reach 100% of possible states (a total of $19 \times 19 = 361$ unique states) in approximately $700$ steps, and 80% of possible states ($\approx 290$ states) in less than $400$ steps. These counts also include the $n_{init}$ number of random steps taken preceding training. Our agent is also able to learn highly interpretable abstract representations in very few environment steps (as shown in Figure 2a) as it explores its state space.

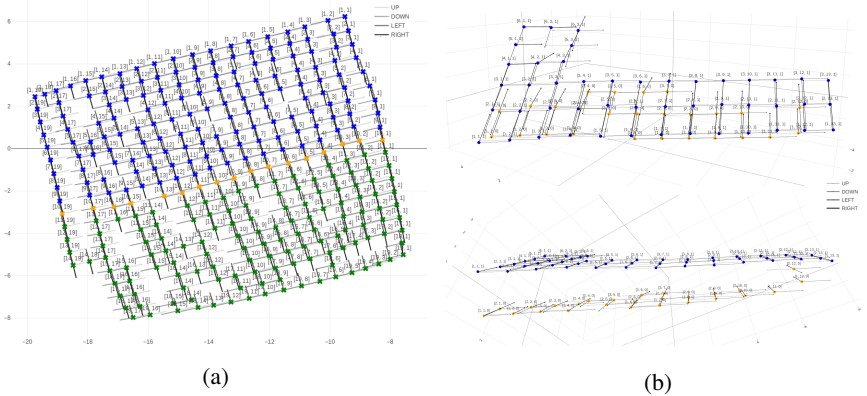

(a)

(b)

Figure 2: (a): A learned abstract representation of the open labyrinth environment from Figure 1a after 500 environment steps. Colors denote which side of the maze the agent was in, grid coordinates and transitions are shown. (b): Two views of the same learned abstract 3-dimensional representation of our multi-step maze after 300 steps. Orange and blue points denote states with and without keys respectively. Our agent is able to disentangle states where the agent has a key ($z = 1$) and when it doesn't ($z = 0$) as seen in the distance between orange and blue states. Meaningful information about the agent position is also maintained in the relative positions of states in abstract state space.

### 6.1.2 4-ROOM LABYRINTH

While the learned dynamics model and abstract representation are able to create an interpretable representation of the open labyrinth environment (Figure 2a), the 4-room labyrinth environment is more challenging. Indeed, our encoder $\hat{e}$ takes a high-dimensional input and compresses it to a low-dimensional representation. In the case of the labyrinth environment, the representation incorporates knowledge related to the position of the agent in 2-dimensions that we call *primary features*. It also learns other information such as agent surroundings (walls, open space) etc., but it does so only via the transition function learned through experience. We call this extraneous but necessary information *secondary features*. As most of these secondary features are encoded only in the dynamics model $\hat{\tau}$, our agent has to experience a transition in order to accurately represent both primary and secondary features.

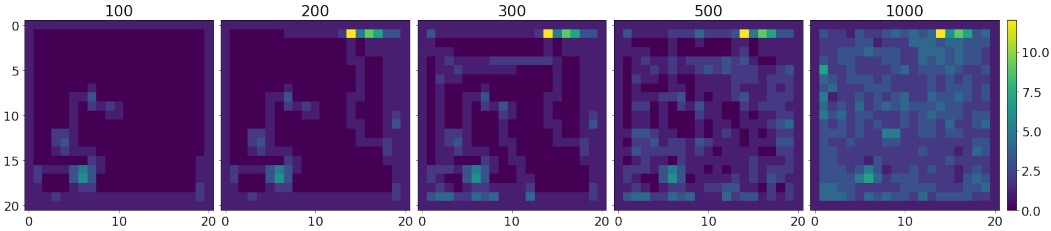

Figure 3: An example of the state counts of our agent in the open labyrinth with $d = 5$ step planning. Titles of each subplot denotes the number of steps taken. The brightness of the points are proportional to the state visitation count.

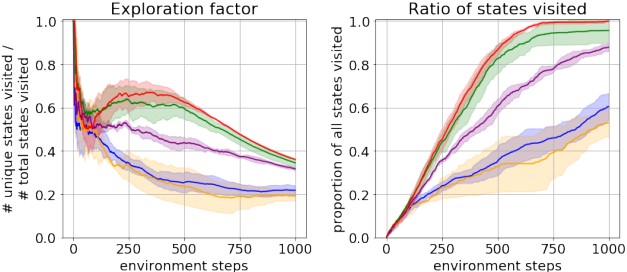

(a) Results for open labyrinth and different variations on policies compared to baselines.

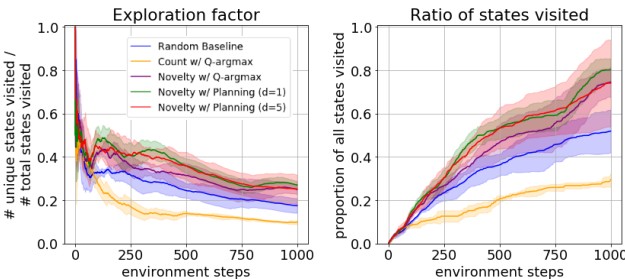

(b) Results for the 4-room labyrinth and different variations on policies compared to baselines.

Figure 4: Labyrinth results for both open labyrinth and 4-room labyrinth over 3 trials.

In this environment specifically, our dynamics model tends to over-generalize for walls between rooms and can thus also fail to try out transitions in the passageways between rooms. Our agent then tends to visit uniformly all the states that are reachable within the known rooms. With an $\epsilon$-greedy policy, our approach still ensures that the agent explores passageways even if it has over-generalized to the surrounding walls.

We run the same experiments as we do on the open labyrinth and report results in Figure 4b. Our approach still outperforms the random and count-based baselines. After visiting most unseen states in its environment, our agent tends to uniformly explore its state space due to the nature of our novelty heuristic. We can see an example of this for the open labyrinth in Figure 3. The bright spot that begins after 200 counts is the agent learning the dynamics of labyrinth walls.

## 6.2 CONTROL AND SUB-GOAL EXPLORATION

In order to test the efficacy of our method on environments with harder dynamics, we conduct experiments on the control-based environment Acrobot (Brockman et al., 2016) and a multi-step maze environment. Our method (with planning depth $d = 5$) is compared to strong exploration baselines with different archetypes:

1. Prediction error incentivized exploration (Stadie et al., 2015)

2. Hash count-based exploration (Tang et al., 2016)

3. Bootstrap DQN (Osband et al., 2016)

In order to maintain consistency in our results, we use the same model architectures and model-free methods throughout. Since we experiment in the deterministic domain, we exclude baselines that require some form of stochasticity or density estimation as baselines (for example, Shyam et al. (2018) and Osband et al. (2017)). Normally, for most approaches that include a model-free component, a step in the environment is taken with every update step. In our experiments, we use orders of magnitude less samples as compared to most model-free RL algorithms (all within the same episode). To ensure a fair comparison between our approach and baselines, we run multiple training iterations in between each environment step for all experiments.

### 6.2.1 ACROBOT

We now test our approach on Acrobot (Brockman et al., 2016), which has a continuous state space unlike the labyrinth environment. We specifically choose this control task because the nature of this environment makes exploration inherently difficult. The agent only has control of the actuator for the inner joint and has to transfer enough energy into the second joint in order to swing it to its goal state. We modify this environment so that each episode is at most 3000 environment steps. We use the default reward setup in Acrobot, where the agent is rewarded with $-1$ at every environment step until it reaches its goal state where it receives a reward of 0. To measure the performance of our exploration approach, we measure the average number of steps that the agent takes to move its second joint above a given line as per Figure 5a in a single episode.

To demonstrate the ability of our method to learn a low dimensional abstract representation from pixel inputs, we use 4 consecutive pixel frames as input instead of the 6-dimensional full state vector. We use a 4-dimensional abstract representation of our state and results from experiments are shown in Table 6.2.1. Our method reaches the goal state more efficiently than our baselines.

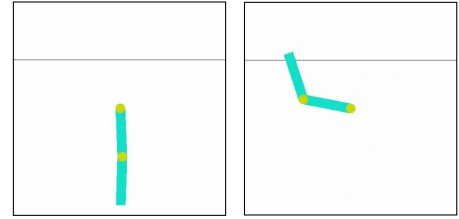
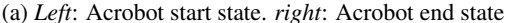
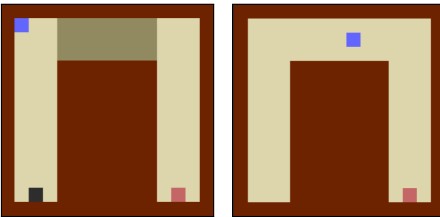

(a) *Left*: Acrobot start state. *right*: Acrobot end state

(b) *Left*: Start of our multi-step maze. *right*: After the agent has collected the key.

Figure 5: Illustrations of the Acrobot and multi-step goal maze environments. *b) Left:* The passageway to the west portion of the environment are blocked before the key (black) is collected. *b) Right:* The passageway is traversable after collecting the key, and the reward (red) is then available. The environment terminates after collecting the reward.

|  | Acrobot | | Multi-step Maze | |
|---|---|---|---|---|
|  | Avg | Stdev | Avg | Stdev |
| Random Policy | 2043.00 | 911.18 | 2416.80 | 1199.16 |
| Prediction error incentivized exploration | 575.25 | 196.89 | 529.50 | 120.59 |
| Hash count-based exploration | 674.25 | 363.59 | 685.50 | 603.24 |
| Bootstrap DQN | 478.00 | 224.48 | 1395.25 | 888.22 |
| Novelty Search with abstract representations | **356.25** | 87.47 | **320.00** | 69.24 |

Table 1: Number of steps necessary to reach the goal state in the Acrobot and the multi-step maze environments (lower is better). Results are averaged over 4 trials. Best results are in bold.

### 6.2.2 MULTI-STEP GOAL MAZE

We also test our method on an environment with a temporal aspect - a maze with the sub-goal of picking up a key to open doors before the main goal of reaching the reward is available. We build our environment with the Pycolab game engine (Stepleton, 2017). The environment can be seen in Figure 5b. Each episode is at most $4000$ environment steps. While this environment does admit an extrinsic reward (1 for picking up the key, 10 for reaching the final state), we ignore these rewards and only focus on intrinsic rewards.

Similarly to the labyrinth environment, the input to our agent is a top-down view of the environment. We also employ an $\epsilon$-greedy policy. From our experiments, we show that our agent is able to learn an interpretable representation of the environment in a sample-efficient manner. It is able to encode this temporal aspect of the environment (whether the key was collected) as shown in Figure 2b. With our intrinsic reward generated with this representation, our agent is able to more efficiently explore its state space and reach the end goal in fewer steps as compared to our baselines.

## 7 RELATED WORK

The proposed exploration strategy falls under the category of directed exploration (Thrun, 1992) that makes use of the past interactions with the environment to guide the discovery of new states. This work is inspired by the Novelty Search algorithm (Lehman and Stanley, 2011) that leverages a nearest-neighbor scoring function, but does so in behavior space. Exploration strategies have been investigated with both model-free and model-based approaches. In Bellemare et al. (2016) and Ostrovski et al. (2017), a model-free algorithm provides the notion of novelty through a pseudo-count from an arbitrary density model that provides an estimate of how many times an action has been taken in similar states.

Several exploration strategies have also used a model of the environment along with planning. Hester and Stone (2012) employ a two-part strategy to calculate intrinsic rewards, combining model uncertainty (from a random-forest based model) and a novelty reward based on $L_1$ distance in feature space. A strategy investigated in Salge et al. (2014); Mohamed and Rezende (2015); Gregor et al. (2016); Chiappa et al. (2017) is to have the agent choose a sequence of actions by planning that leads to a representation of state as different as possible to the current state. In Pathak et al. (2017); Haber et al. (2018), the agent optimizes both a model of its environment and a separate model that predicts the error/uncertainty of its own model. Burda et al. (2018a) similarly uses an intrinsic reward based on the uncertainty of its dynamics model. The agent can thus seek actions that adversarially challenge its knowledge of the environment (Savinov et al., 2018). In Shyam et al. (2018), multiple model forward models of the environment are also employed to plan to observe novel states by using a measure of novelty derived from disagreement between future states.

## 8 DISCUSSION

In this paper, we show that with an interpretable abstract representation of states, our novelty metric is able to serve as an intrinsic reward that enables efficient exploration. By using this novelty metric with a combination of model-based and model-free approaches for planning, we demonstrate the efficiency of our method in multiple environments.

As with most methods, our approach also has limitations. While the problem of distance metrics in high-dimensional space is partially solved in our method with the dimensionality reduction of observations by our encoder, the $\ell_2$-norm still requires a low dimension to be useful (Aggarwal et al., 2002). This implies that our novelty metric may lose its effectiveness as we increase the dimension of our abstract representation.

In addition, our exploration strategy benefits greatly from the meaningful abstractions and internal model. In some cases, the model can over-generalize with the consequence that the low-dimensional representation loses information that is crucial for the exploration of the entire state space. An interesting direction for future work would be find ways of incorporating the secondary features mentioned in Section 6.1.2.

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

## A    DISCUSSION ON THE ENTROPY CONSTRAINT

As for our soft constraints on representation magnitude, we use a local constraint instead of a global constraint on magnitude such that it is more suited for our novelty metric. If we are to calculate some form of intrinsic reward based on distance between neighboring states, then this distance needs to be non-zero and ideally consistent as the number of states in our history increases. In the global constraint case, if the intrinsic rewards decreases with an increase in number of states in the agent's history, then the agent will fail to be motivated to explore further. Even though the entropy maximization losses ensures the maximization of distances between random states, if we have $|H_s|$ number of states in the history of the agent, then a global constraint on representation magnitude might lead to

$$\lim_{|H_s| \to \infty} \mathbb{E}_{(s,s') \sim (H_s, H_s)}[\|s - s'\|_2] = 0. \tag{9}$$

## B    MOTIVATION FOR RANKED WEIGHTING OF K-NN SCORES

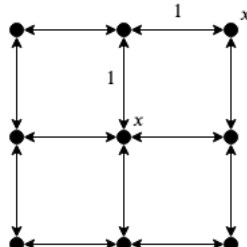

Figure 6: Illustration of a simplistic $3 \times 3$ grid-like environment, with $\mathcal{A} = \{\text{up}, \text{down}, \text{left}, \text{right}\}$ and distance 1 between neighboring states. All actions which lead out of the grid are no-ops.

Here $H_s$ is a list of all possible states in $\mathcal{S}$ in any order, with each possible state appearing only once. If we let $k = 4$, we have that:

$$\rho_{\mathcal{X}}(x', \hat{e}(H_s)) = \frac{0 + 1 + 1 + \sqrt{2}}{4} = \frac{2 + \sqrt{2}}{4} \tag{10}$$

$$\rho_{\mathcal{X}}(x, \hat{e}(H_s)) = \frac{0 + 1 + 1 + 1}{4} = \frac{3}{4} \tag{11}$$

$$\tag{12}$$

While it may seem redundant to include the 1st nearest neighbor distance in this metric (which would be itself if we've visited the state), the 1st nearest neighbor is non-zero when we calculate the novelty of a predicted state using our learned transition function $\hat{\tau}$. From this example, we can see that there is a bias towards states with fewer direct neighbors due to the nature of our novelty metric. This poses an issue - if our goal is for sample-efficient exploration of our state space, then there is no reason to favor states with less direct neighbors.

## C    MOTIVATION FOR USING $L_2$ NORM AS A DISTANCE MEASURE IN ABSTRACT REPRESENTATION SPACE

We consider the entropy maximization loss $L_{d1}$ and consecutive distance loss $L_{consec}$. On the one hand, the entropy maximization loss ensures maximal $\ell_2$ distance between two random states due to the inverse relationship of the negative exponential. On the other hand, minimization of our consecutive distance loss ensures that the distance between two consecutive states is $\leq \omega$. We consider the minima of the combination of these two losses for a consecutive states $s$ and $s'$:

$$L* = \min_{\theta_{\hat{e}}} L_{d1}(\theta_{\hat{e}}) + L_{consec}(\theta_{\hat{e}})$$
$$= \min_{\theta_{\hat{e}}} exp(-C_d \|\hat{e}(s; \theta_{\hat{e}}) - \hat{e}(s'; \theta_{\hat{e}})\|_2) + max(\|\hat{e}(s; \theta_e) - \hat{e}(s'; \theta_e)\|_2 - \omega, 0)$$

The minimum of the combination of the losses is obtained when the distance between $s$ and $s'$ is $\omega$. These losses shape abstract state space so that $\ell_2$ norm as a distance measure encodes a notion of closeness in state space that we leverage in our novelty heuristic.

# D  LIMITING BEHAVIOR FOR NOVELTY HEURISTIC

In the limiting case, our novelty heuristic is 0 for all states. Indded, if we consider the case where each state has been visited $k$ times, the $k$ nearest neighbors to state $s$ are all identical and equal to $s$ in the deterministic case. Based on equation 8, we have the following novelty heuristic score for each state:

$$\rho_{\mathcal{X}}\left(\hat{e}(s;\theta_{\hat{e}})\right) = \sum_{i=1}^{k}\frac{1}{i}dist\left(\hat{e}(s;\theta_{\hat{e}}),\hat{e}(s;\theta_{\hat{e}})\right) = 0$$

As for stochastic environments, we let $s''$ be a stochastic observation of the state $s$, and have that

$$\mathbb{E}[\rho_{\mathcal{X}}\left(\hat{e}(s;\theta_{\hat{e}})\right)] = \sum_{i=1}^{k}\frac{1}{i}\mathbb{E}[dist\left(\hat{e}(s;\theta_{\hat{e}}),\hat{e}(s'';\theta_{\hat{e}})\right)].$$

If $\mathbb{E}[dist\left(\hat{e}(s;\theta_{\hat{e}}),\hat{e}(s'';\theta_{\hat{e}})\right)] \to 0$ as the encoder reaches optimality, then our novelty heuristic will return 0 in expectation in the limiting case.

# E  EXPERIMENTAL SETUP AND HYPERPARAMETERS

For all of our experiments, we use a batch size of 64 and take 64 random steps transitions before beginning training. We also use the same discount factor for all experiments ($\gamma = 0.8$) and the same freeze interval for target parameters 1000. The reason behind our low discount factor is due to the high density of non-stationary intrinsic rewards in our state. For all model-based abstract representation training, the following hyperparameters were all kept constant: minimum distance between consecutive states $\omega = 0.5$, slack ratio $\delta = 6$ and transition model dropout of $0.1$. For all experiments run with our novelty metric, we use $k = 5$ for our k-NN calculations.

## E.1  NEURAL NETWORK ARCHITECTURES

For reference, 'Dense' implies a full-connected layer. 'Conv2D' refers to a 2D convolutional layer with stride 1. 'MaxPooling2D' refers to a max pooling operation. All networks were trained with the RMSProp optimizer. Throughout all experiments, we use the following neural network architectures:

### E.1.1  ENCODER

For all our non-control task inputs, we flatten our input and use the following feed-forward neural network architecture for $\hat{e}$:

- Dense(200, activation='tanh')
- Dense(100, activation='tanh')
- Dense(50, activation='tanh')
- Dense(10, activation='tanh')
- Dense(abstract representation dimension).

For our control task, we use a convolution-based encoder:

- Conv2D(channels=8, kernel=(3,3), activation='tanh')
- Conv2D(channels=16, kernel=(3,3), activation='tanh')
- MaxPool2D(pool size=(4,4))
- Conv2D(channels=32, kernel=(3,3), activation='tanh')

- MaxPool2D(pool size=(3,3))

- Dense(abstract state representation dimension).

### E.1.2 TRANSITION MODEL

The input to our transition model is a concatenation of an abstract representation and an action. We use the following architecture

- Dense(10, activation='tanh', dropout=0.1)

- Dense(30, activation='tanh', dropout=0.1)

- Dense(30, activation='tanh', dropout=0.1)

- Dense(10, activation='tanh', dropout=0.1)

- Dense(abstract representation dimension)

and add the output of this to the input abstract representation.

### E.1.3 REWARD AND DISCOUNT FACTOR MODELS

For both reward and discount factor estimators, we use the following architecture:

- Dense(10, activation='tanh')

- Dense(50, activation='tanh')

- Dense(20, activation='tanh')

- Dense(1).

### E.1.4 Q FUNCTION APPROXIMATOR

We use two different architecture based on the type of input. If we use the concatenation of abstract representation and action, we use the following architecture:

- Dense(20, activation='relu')

- Dense(50, activation='relu')

- Dense(20, activation='relu')

- Dense($n_{actions}$)

For the pixel frame inputs for our control environments, we use:

- Conv2D(channels=8, kernel=(3,3), activation='tanh')

- Conv2D(channels=16, kernel=(3,3), activation='tanh')

- MaxPool2D(pool size=(4,4))

- Conv2D(channels=32, kernel=(3,3), activation='tanh')

- MaxPool2D(pool size=(3,3))

- Dense($n_{actions}$).

Finally, for our (purely model free) gridworld environments we use:

- Dense(500, activation='tanh')

- Dense(200, activation='tanh')

- Dense(50, activation='tanh')

- Dense(10, activation='tanh')

- Dense($n_{actions}$)

As for our Bootstrap DQN implementation, we use the same architecture as above, except we replace the final Dense layer with 10 separate heads (each a Dense layer with $n_{actions}$ nodes).

### E.2 LABYRINTH ENVIRONMENTS

Both environments used the same hyperparameters except for two: we add an $\epsilon$-greedy ($\epsilon = 0.2$) policy for the 4-room maze, and increased $n_{freq}$ from 1 to 3 in the 4-room case due to unnecessary over-training. We have the following hyperparameters for our two labyrinth environments:

- $n_{iters} = 30000$
- $\alpha = 0.00025$

### E.3 CONTROL ENVIRONMENT

In our Acrobot environment, the input to our agent is 4 stacked consecutive pixel frames, where we reduce each frame down to a $32 \times 32$ pixel frame. Our abstract representation dimension is 4. We use a learning rate of $\alpha = 0.00025$ for all experiments. We train for $n_{iters} = 30000$ for both model-free experiments and $n_{iters} = 50000$ for both experiments incorporating model-based components - this discrepancy is due to the need for more iterations for the model-based portion to converge.

### E.4 MULTI-STEP MAZE ENVIRONMENT

In our multistep maze environment, the input to our agent is a single $15 \times 15$ frame of an overview of the environment. Our abstract representation dimension is 3. We use an $\epsilon$-greedy ($\epsilon = 0.1$) policy for this environment. We use $\alpha = 0.00025, n_{iters} = 30000$ for our model-free algorithms and $\alpha = 0.000025, n_{iters} = 50000$ for experiments that include a model-base component. This is again due to the need for our model-based component to converge.

## F ABLATION STUDY

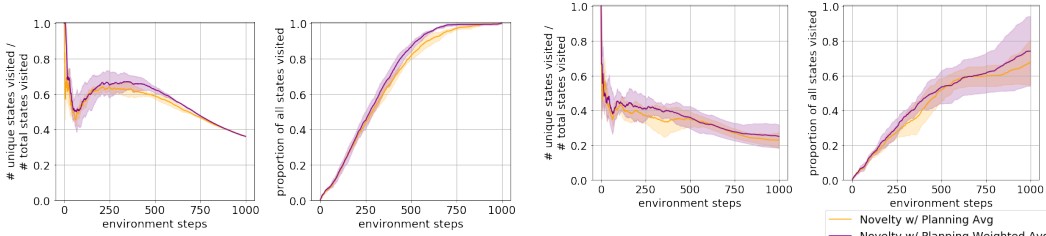

(a) Results for open labyrinth between unweighted and weighted k-NN averages.

(b) Results for the 4-room labyrinth between unweighted and weighted k-NN averages.

Figure 7: Ablation results for unweighted versus weighted nearest neighbor heuristics (over 3 trials).

We perform an ablation study between unweighted (orange) and weighted (red) nearest neighbor heuristics on the open labyrinth (left) and 4-room labyrinth (right) environments. While the weighted nearest neighbor approach does improve the bias towards visiting more fewer-neighbor states, the improvements are relatively marginal.

