# OpenReview forum: "Novelty Search in representational space for sample efficient exploration"
_ICLR.cc/2020/Conference — Reject_

### Official Review · AnonReviewer1 · 2019-10-22
**Official Blind Review #1**

**Rating:** 3

**Review:**

This paper proposes a method for efficient exploration in tabular MDPs as well as a simple control environment. The proposed method uses deterministic encoders to learn a low dimensional representation of the environment dynamics, which preserves distances between states such that “close” states in the full MDP are close in the learned representation. An intrinsic reward is formulated based on a measure of novelty, given by distance between new states, and a stored replay buffer.  Along with the dynamics model, a model-free agent employs Q learning to find a good policy. Experiments are performed on 3 tabular environments and the acrobot control task.

Pros:
1.	Overall the paper is clear and the proposed method makes sense intuitively. The intrinsic reward is cheap to compute and the state abstraction offers a nice way to visualize state differences the agent thinks are important.

2.	The method seems to be sample efficient with regard to strong baselines like [1]

Cons:
1.	It seems difficult to argue the efficacy of a low-dimensional state representation that doesn’t scale with state dimensionality. As shown in [2] and [3], learning effective state abstractions in high dimensions can require considerably more effort.

2.	Given that there exist novelty based intrinsic rewards which compute state abstractions in high dimensional environments [3], I find it hard to see the usefulness of the proposed method.

3.	The choice of distance metric for the representational space is not well motivated. As correctly stated by the authors, the L_2 norm will cease to be a good metric as state dimensionality increases.

4.	There are too many grid-world experiments. The point of the first two experiments can be made simply using the four-room environment. This could make room for a more interesting experiment such as MuJoCo Ant Maze.

My main issue with the work in its current form is that the method is too light in terms of technical contribution. Simple methods are ok (even valuable!) but there should be a certain about of rigorous analysis which shows that the simple method can be used as a foundation for further work. For a method which mostly examines tabular environments, I expect some analysis of the methods efficiency with regard to data efficiency -- the main point of the paper. [1] and [4] which are used as comparisons, provide such analysis. If a convincing theoretical analysis is out of reach, then it could be sufficient to provide extensive experimental evidence supporting the claims. In this case it could include an examination of different metrics, additional (ideally more difficult) environments, and comparison to other baselines like [5], [3].

Due to what I see as a lack of technical contribution, I do not recommend acceptance to ICLR at this time.

A more compelling submission would include the following:
●	A more detailed motivation for why the L2 norm makes sense as a distance metric.
○	In an abstract space, it's more natural to use a statistical distance like the KL or JS divergence. These metrics have drawbacks, but they should be discussed
●	An analysis of the limit behavior of the proposed method. Given enough time an intrinsic reward should explore every state in a deterministic environment. Does this happen in the limiting case -- if not, is the margin acceptable.
●	More extensive experiments. This method can clearly admit convolutional architectures so experiments on more interesting environments are viable. Though I believe this would require more complex models such as a VAE, and may change the submission considerably.

Minor notes
●	Section 3: “when [the distance between transitions is less than the slack ratio] the transitions are mostly accurate within a ball of radius \frac{w}/{\delta}. This is too vague, what does mostly accurate mean?
●	Eq (6), is \alpha a hyperparameter as well as the learning rate? If \alpha is just the learning rate than the equation is incorrect, because the learning rate is applied to the gradient of the loss, not the loss itself.
●	The description of the planning algorithm and Q learning in section 4 is a little sloppy, a clearer description would be appreciated.
●	Computing novelty with respect to a state’s nearest neighbors is problematic at scale. This point should be at least acknowledged.


[1] Osband, Ian, et al. "Deep exploration via bootstrapped DQN." Advances in neural information processing systems. 2016.
[2] Kim, Hyoungseok, et al. "EMI: Exploration with Mutual Information." International Conference on Machine Learning. 2019.
[3] Ha, David, and Jürgen Schmidhuber. "World models." arXiv preprint arXiv:1803.10122 (2018).
[4] Bellemare, Marc, et al. "Unifying count-based exploration and intrinsic motivation." Advances in Neural Information Processing Systems. 2016.
[5] Pathak, Deepak, et al. "Curiosity-driven exploration by self-supervised prediction." Proceedings of the IEEE Conference on Computer Vision and Pattern Recognition Workshops. 2017.


**Experience Assessment:**

I have read many papers in this area.

**Review Assessment: Checking Correctness Of Derivations And Theory:**

I assessed the sensibility of the derivations and theory.

**Review Assessment: Checking Correctness Of Experiments:**

I carefully checked the experiments.

**Review Assessment: Thoroughness In Paper Reading:**

I read the paper thoroughly.

---

> ### Author Response · Authors · 2019-11-13
> **Rebuttal for review #1**
>
> Thank you for the feedback. We'll try to address some of the issues you brought up here:
>
> 2.    Given that there exist novelty based intrinsic rewards which compute state abstractions in high dimensional environments [3], I find it hard to see the usefulness of the proposed method.
>
> While intrinsic rewards do exist for high-dimensional environments, our approach specifically leverages low-dimensional representational space that does not solely rely on frame prediction as per [3]. There has been evidence (Appendix B.1 from Lavet et al. 2018) that using some form of reconstruction is too strong of a constraint to build meaningful representations in lower dimensions. Our approach leverages the fact that distance in our abstract space has significance in terms of the “novelty” of a state.
>
> ●	A more detailed motivation for why the L2 norm makes sense as a distance metric.
>
> We’ll try and add something in the appendix motivating our use of L2 norm.
>
> ●	In an abstract space, it's more natural to use a statistical distance like the KL or JS divergence. These metrics have drawbacks, but they should be discussed
>
> Since each abstract state is not constrained to have an L2 norm <= 1, I don’t see how probability distances could serve as a replacement to L2 norm. If there's any literature demonstrating the use of distance metrics similar to KL or JS divergence for non-probability measures we'd be open to considering this.
>
> ●	An analysis of the limit behavior of the proposed method. Given enough time an intrinsic reward should explore every state in a deterministic environment. Does this happen in the limiting case -- if not, is the margin acceptable.
> As per our comment to reviewer 2, we’ll try and add some analysis for limiting cases.
>
> ●	More extensive experiments. This method can clearly admit convolutional architectures so experiments on more interesting environments are viable. Though I believe this would require more complex models such as a VAE, and may change the submission considerably.
>
> We agree that this approach could be used in more extensive environments and approaches. But specifically for a VAE approach, a reconstruction loss imposes a pixel-similarity constraint between input and output which distorts our low-dimensional representations.
>
> As for the minor notes, we'll try to address them accordingly in the coming edit.

---

### Official Review · AnonReviewer3 · 2019-10-22
**Official Blind Review #3**

**Rating:** 6

**Review:**

This paper proposes a method of sample-efficient exploration for RL agent. The main problem at hand is the presence of irrelevant information in raw observations. To solve this problem, the authors leverage novelty heuristics in a lower-dimensional representation of a state, for which they propose a novelty measure. Then they describe a combination of model-based and model-free approaches with the novelty metric used as an intrinsic reward for planning that they use to compare with baselines solutions. They conduct experiments to show that their algorithm outperforms random and count-based baselines. They show that their approach has better results then Random Policy, Prediction error incentivized exploration, Hash count-based exploration, Bootstrap DQN while playing Acrobot and Multi-step Maze.

Authors propose a novel approach to the problem of exploration. They test their method by experiments conducted in two environments, where they use the same model architectures and model-free methods for all types of novelty metrics, which shows the contribution of the proposed method in the results of learning.

To sum up, the decision is to accept the paper as the problem is important, ideas are rather new, and results are better compared to other approaches.

1. The dependence of the quality of the dimensionality representational state is unclear. For different environments, different abstract representation dimensions are chosen, but the reason is not explained.
2. Word "we" is overused in the article

**Experience Assessment:**

I do not know much about this area.

**Review Assessment: Checking Correctness Of Derivations And Theory:**

N/A

**Review Assessment: Checking Correctness Of Experiments:**

I assessed the sensibility of the experiments.

**Review Assessment: Thoroughness In Paper Reading:**

I read the paper at least twice and used my best judgement in assessing the paper.

---

### Official Review · AnonReviewer2 · 2019-10-23
**Official Blind Review #2**

**Rating:** 1

**Review:**

The paper proposes an approach to exploration by utilizing an intrinsic reward based on distances in a learned, evolving abstract representation space. The abstract space is learned utilizing both model-free and model-based losses, and the behaviour policy is based on planning combining the model-free and model-based components with an epsilon-greedy exploration strategy. Learning the abstract representation space itself is based on a previous work, but the contribution of this paper is the utility of it to design the reward bonus for exploration by utilizing distances in this evolving representation space.

As it stands, I am leaning towards rejecting the paper, for the following reasons.
(1) while the idea proposed is interesting, the current work rather explores it in a limited manner which is unsatisfactory.
(2) I think the presentation of the bonus itself -- novelty search (Section 4), which is the core of the paper, is rather unclear. (3) The assumption of deterministic transition dynamics may be ignored in favour of games which seem to be our benchmarks, but the results presented for the control tasks, Table 1, are not statistically significant, and the paper is missing details about the architecture/sweep for the baselines experimented with.
(4) Parts of the paper is rather unclear/feels disconnected -- for instance, the interpretable abstract representation bit; this was a loss in the original work, and seems to be just mentioned arbitrarily here while the loss isn't really used (unless it is used, and not mentioned in the paper).
(5) Overall, the proposed reward bonus is a heuristic whose specific design choice isn't statistically shown to be useful (Ablation in Appendix), and the empirical results comparing to other methods are underwhelming.

Here are my main points of concern which I hope the authors address in the rebuttal:
(1) Designing reward bonuses to induce exploratory behaviour in the agent has seen a surge of publications in the Deep RL literature in recent years. The key property all these methods aim for is a bonus that pushes the agent to the boundaries of its current "known region", and then rely on the stochasticity due to epsilon-greedy to cross that boundary -- pushing this boundary further. While this is different from exploration to reduce uncertainty, it is nonetheless a reasonable approach leading to competitive policies when evaluated in deep RL. But a characteristic all these bonuses aim for is that they fade away with time -- for instance count-based bonus are inversely proportional to visit counts, or prediction error bonuses go to 0 as the prediction becomes more accurate. But what do these novelty bonuses converge to? Is it just a stationary value based on consecutive loss parameter (in which case the hope is they don't affect the external reward scale, they just shift it uniformly)?
(2) What exactly are the nearest neighbours? Is it a search based on the data in the buffer or is it a notion of temporal neighbours?
(3) If it's temporal, why would there ever be biased for some states -- "We do this in order..novel states".
(4) I was completely unable to understand the section in the Appendix which is making a case for the ranked weighting. If you have a succinct explanation for the heuristic it'd be great.
(5) Further, as a heuristic it is mentioned that l2 norm may not be effective if the dimensionality of the representation space is increased. So why the heuristic? I think it either needs more empirical validation, or a theoretical justification.
(6) While the evaluation scheme used in the paper to quantify the exploration of the behaviour policy is interesting -- y-axis of plots in Figure 4 for the Labyrinth task -- why/what exactly is the role of Figure 2? Is the interpretability loss used here? Is it to reason for utilizing e-greedy instead of a purely-greedy behaviour? I think this is a little unclear, and can be better clarified. Further, the distinction of primary and secondary features is interesting, but their clear demarcation is rather questionable in more complicated domains -- in the abstract space.
(7) Do you have a hypothesis for why the 1-step value functions are not sufficient for decision making in this simple domain -- labyrinth - with the abstract representations?
(8) If model-based algorithms get more steps to learn shouldn't model-free too? I'm not sure I understand the reasoning for the experiment design choice.
(9) Whats the architecture used for Bootstrap DQN? It needs to have multiple heads -- but based on the current architecture that doesn't seem likely.
(10) Are the extrinsic rewards ignored in learning -- "only focus on intrinsic rewards" (Section 6.2.2)? If they are for the proposed method, are they for the competitors too? If so why, and what is the reward for Bootstrap DQN?
(11) I think the Discussion section raises interesting points about interpretability and metric learning, but I do think the conclusions drawn are a little inflated.
(12) The ablation study in Section D of the Appendix is not statistically significant -- so why is wighted reward useful? Please comment.
(13) How would stochasticity in transition dynamics affect the abstract representation space? Discussing this would be very interesting.
(14) Learning curves for the control tasks?

Comments about typos/possible points of confusion:
(1) The last para in Section 6.1 -- discusses "open" labyrinth heat map, then what do we mean by learning the dynamics of the wall? There is no wall in open, right?
(2) In Section 4 -- I think x_{t+1} is an estimate from the unrolled model -- \hat{x}_{t+1}? Further, it would be helpful to mention that it is an estimate based on the learned model.
(3) n_freq is used in the pseudocode in the main paper -- but no mention of it to explain it is made in the main.
(4) Contrasting the work to existing literature would be useful (in the Related Work section; as opposed to summarizing existing work).
(5) buffered Q network --> target networks?


**Experience Assessment:**

I have published one or two papers in this area.

**Review Assessment: Checking Correctness Of Derivations And Theory:**

I carefully checked the derivations and theory.

**Review Assessment: Checking Correctness Of Experiments:**

I carefully checked the experiments.

**Review Assessment: Thoroughness In Paper Reading:**

I read the paper thoroughly.

---

> ### Author Response · Authors · 2019-11-13
> **Rebuttal for review #2**
>
> Thank you for the feedback. Here are our responses to the points brought forward:
>
> (1) Our novelty bonuses exhibit similar asymptotic behavior as count-based rewards. One way of seeing them is as a count-based reward that takes into account locality of surrounding states. Similar to count-based rewards, our novelty-based rewards will also converge to 0 as we visit enough states (specifically, our novelty bonus will return 0 once we visit the given state > k times). Since we're only considering distances from the first k neighbors for our novelty heuristic, once we visit the same state k times our novelty heuristic returns 0. We could definitely add a short section on limiting behavior in the appendix of our reward if that will clarify things.
>
> (2) Nearest neighbours is based on L2 distance between the abstract representation of the observation in question and abstract representation of observations in the data buffer as per equation (8).
>
> (4) In the graph in Figure 6, each node represents a state in abstract representational space (R^2 in this case). In this toy example, each of the 9 states have been visited once. We calculate the (unweighted) nearest neighbors scores for the states x’ and x. We see that due to the structure of the state space, our novelty reward will be higher in states with less direct neighbors (x’) as compared to states with many neighbors (x). We’ll further clarify these points in the paper.
>
> (5) We’re not sure we understand this question. We justify the use of L2 norm because the representation space is in (significantly) lower dimension as compared to its observations.
>
> (6) Interpretability loss is not used in any of the experiments. Figure 2 is to illustrate that our method is able to learn an interpretable representation of our state in lower dimensions in the low sample setting. We do not mention the interpretability loss because it plays no role in our experiments. The confusion may be coming from the fact that we call our representations "interpretable" - we mean interpretable in the literal sense.
>
> (7) Our experiments show that the 1-step value function is able to do well in the simple labyrinth example, but has higher variance due to the learnt Q function (Figure 4a).
>
> (8) We assign fewer training iterations to our model-free methods because they converge after fewer iterations. We originally tried the same number of iterations for both, but realized the faster convergence for model-free methods means wasted iterations because of convergence.
>
> (9) We’ll add further clarifications for our Bootstrap DQN architecture - yes, we do use multiple heads (as described by the algorithm).
>
> (10) Yes, they’re ignored for all of our methods. Bootstrap DQN is still able to explore due to the variances of it’s Q estimates due to the multiple heads.
>
> (11) Could you direct us to which conclusions seem inflated?
>
> (12) As per Appendix B, while there may not be a significant difference in the open Gridworld example, bias issues may arise in environments with more complicated domains. We decided to use the weighted nearest neighbors in order to reduce this bias.
>
> (13) While we don't mention anything about stochasticity in the discussion, we do briefly mention the possibility of using generative methods for stochastic environments. While this is a very interesting point, we decided to consider this in future work.
>
> (14) We will try to include a few of the curves for our losses for Acrobot in our appendix.
>
> As for points of confusion/typos, we'll look to clarify the points you brought forward.

---

> > ### Comment · AnonReviewer2 · 2019-11-14
> > **Response to rebuttal; clarity issues**
> >
> > Thank you for your response.
> >
> > (1,2) I genuinely fail to see why the bonus would converge to 0. I think it would converge to omega -- the constraint enforced for successive embeddings. But in any case, finding k-nearest neighbours seems sensitive to the contents of the replay buffer. What is the size of the buffer in the current experiments? This is crucial and not mentioned.
> >
> > (4) But the distances are based on the abstract space embeddings, which reflect the dynamics. If we consider all the states as being in the buffer, presumably the avg. distance wrt x' should be more than the average distance to x. This is not a result of the neighbours -- but rather the dynamics of the problem. I don't see why that is an issue or how the rank based weighting alleviates it. For equation 10 and 11 if we used rank based weighting x' would still have a higher bonus -- sqrt(2)/4 for x' vs. 1/4 for x.
> >
> > Actually, now that I read the paragraph below equation (12) in the appendix -- it seems like the nearest neighbours aren't a search, but are rather a simulation from the transition model in abstract space ("the 1st nearest neighbour is non-zero when we calculate the novelty of a predicted state using our learned transition function). So what are the nearest neighbours?
> >
> > (5) The work acknowledges that l2 norm in the abstract space may not be informative if the dimensionality is big -- but the current experiments do not make a comprehensive pitch for the proposed heuristic reward bonus scheme even in the low dimensionality regime. Therefore it is possible that the proposal is fundamentally flawed -- even if we do not consider l2 norm as being the bottleneck for extending the proposal to a higher dimensional space -- or needs better empirical evidence.
> >
> > (7) Sorry I missed this -- thank you for pointing it out. Although I don't think the variance is high comparatively -- its learning is just poor.
> >
> > (8) I'm not sure what it means for model-free to have converged; presumably, it is still improving/exploring? So the difference in number of steps seems rather not justified.
> >
> > (10) Ignoring extrinsic reward: why/how does that lead to a policy that does well when evaluated -- even for the policy learned with the abstract space? Bootstrap DQN would still explore sure -- but it would be exploring based on the variance estimates due to its initialization. Why is this a legitimate adaptation of the algorithm?
> >
> > (12) This is a hypothesis that has not been validated empirically. As the main contribution of the paper is the design of the reward bonus -- I really think the work needs to evaluate the proposal more concretely.
> >
> > (13) Learning curves for both Acrobot and Multi-step Goal Maze would be useful. Also, I do not mean the loss curves -- I mean the evaluation curves, over the learning period.
> >
> > Answers to these queries would be clarifying to me personally, but in any case, the proposed exploration bonus is unclear, and limitedly explored. The comparison to baselines is scarce. I do not think the paper can be accepted in its current form. Clarifying it would require another peer review of the manuscript, and therefore, I'm changing my score to a reject. I hope the comments are useful, and the future drafts are clear and thorough.

---

### Decision · Program_Chairs · 2019-12-19

**Decision:**

Reject

**Comment:**

The two most experienced reviewers recommended the paper be rejected.  The submission lacks technical depth, which calls the significance of the contribution into question.  This work would be greatly strengthened by a theoretical justification of the proposed approach.  The reviewers also criticized the quality of the exposition, noting that key parts of the presentation was unclear.  The experimental evaluation was not considered to be sufficiently convincing.  The review comments should be able to help the authors strengthen this work.